# A test of affect processing bias in response to affect regulation

**Keith A. Bush** [ID]*, **Clinton D. Kilts**

Brain Imaging Research Center, Department of Psychiatry, University of Arkansas for Medical Sciences, Little Rock, Arkansas, United States of America

* kabush@uams.edu

**Data Availability Statement:** The authors have made a Brain Imaging Data Structure (BIDS) formatted variant of the full study dataset publicly available (as well as raw real-time log files and training materials) via the Open Science

## Abstract

In this study we merged methods from machine learning and human neuroimaging to test the role of self-induced affect processing states in biasing the affect processing of subsequent image stimuli. To test this relationship we developed a novel paradigm in which (n = 40) healthy adult participants observed affective neural decodings of their real-time functional magnetic resonance image (rtfMRI) responses as feedback to guide explicit regulation of their brain (and corollary affect processing) state towards a positive valence goal state. By this method individual differences in affect regulation ability were controlled. Attaining this brain-affect goal state triggered the presentation of pseudo-randomly selected affectively congruent (positive valence) or incongruent (negative valence) image stimuli drawn from the International Affective Picture Set. Separately, subjects passively viewed randomly triggered positively and negatively valent image stimuli during fMRI acquisition. Multivariate neural decodings of the affect processing induced by these stimuli were modeled using the task trial type (state- versus randomly-triggered) as the fixed-effect of a general linear mixed-effects model. Random effects were modeled subject-wise. We found that self-induction of a positive valence brain state significantly positively biased valence processing of subsequent stimuli. As a manipulation check, we validated affect processing state induction achieved by the image stimuli using independent psychophysiological response measures of hedonic valence and autonomic arousal. We also validated the predictive fidelity of the trained neural decoding models using brain states induced by an out-of-sample set of image stimuli. Beyond its contribution to our understanding of the neural mechanisms that bias affect processing, this work demonstrated the viability of novel experimental paradigms triggered by pre-defined cognitive states. This line of individual differences research potentially provides neuroimaging scientists with a valuable tool for exploring the roles and identities of intrinsic cognitive processing mechanisms that shape our perceptual processing of sensory stimuli.

## Introduction

Our capacity to process and regulate emotions is central to our ability to optimize psychosocial functioning and quality of life [1]. As a corollary, disruptions in emotion processing and

Framework: https://osf.io/yn4vq/. The authors have made the full source code used in this analysis publicly available: https://github.com/kabush/CTER. The source code used to convert raw data files to BIDS format has also been made publicly available: https://github.com/kabush/CTER2bids. All other relevant data is contained within the manuscript or its supporting files.

**Funding:** This study was funded by Brain and Behavior Research Foundation NARSAD Young Investigator Award #26079 sponsored by the Families for Borderline Personality Disorder Research (K.A.B). Elements of the real-time fMRI infrastructure deployed in this work were supported by National Science Foundation grant BCS-1735820 (K.A.B). Additional personnel support was provided by National Institute on Drug Abuse grant 1T32DA022981 (C.D.K). Subject recruitment for the project was supported by the UAMS Translational Research Institute (TRI) through the National Center for Advancing Translational Sciences (1U54TR001629-01A1). The funders had no role in study design, data collection and analysis, decision to publish, or preparation of the manuscript.

**Competing interests:** The authors have declared that no competing interests exist.

regulation are broadly ascribed to psychiatric illnesses including borderline personality disorder, depression, anxiety disorders, PTSD, and substance-use disorders [2] which negatively impact quality of life and functioning [3, 4]. In light of this, scientists and clinicians seek to both develop and understand mental strategies that volitionally reduce negatively biased emotional states. Neuroimaging, in particular, has provided critical insight into the functional neurocircuits involved in efficacious emotion regulation strategies [5, 6]. However, the basic neurobiological mechanisms by which mental strategies induce adaptive emotion processing over time remain elusive.

Research into the effects of temporal context on affect and emotion processing may have implications for increasing our understanding of the neural bases of emotion regulation. Prior work has demonstrated that changing affective context prior to an emotional target shapes the processing of that target. Such priming effects both accelerate and weaken the emotional response to affectively congruent target stimuli [7]. Manipulations of affect processing state impact the temporal structure of the neural responses to subsequent affective image stimuli [8] as well as the corollary psychophysiological responses to those stimuli [9, 10]. Further, stimulus-cued emotion processing states bias the self-reported perception of successive emotional stimuli [11].

These findings are consistent with effects that would be predicted by the deployment of situational and attentional modification strategies according to the process model of emotion regulation [12] and point to potential mechanisms underlying emotion regulation-related changes to emotion processing. However, the neural representation of the observed ability of affective cognitions related to these strategies to bias subsequent emotional responses has not yet been tested. Thus, the primary aim of this work was to contribute to our knowledge of the mechanisms underlying emotion regulation (operationalized as affect regulation) by experimentally demonstrating that self-induced and verified affect processing states bias the affect processing of subsequent image stimuli.

Real-time functional magnetic resonance imaging (rtfMRI), when used to generate brain activation feedback [13] (i.e., rtfMRI-guided neuromodulation or neurofeedback), reflects a promising methodology that has not to our knowledge been applied for mechanistic testing of how the neural correlates of such feedback-induced affect processing states bias subsequent affect processing. Here, the applied advantage of rtfMRI is that self-induced neurocognitive states (achieved via rtfMRI guidance) can be verified and used as independent experimental variables to trigger subsequent affective stimulus-response characterizations. Yet, a challenge to rtfMRI-guided neuromodulation studies, and brain computer interface (BCI) research in general, is the large individual variation observed in subjects' ability to volitionally modulate their cognitive states–the well-known "BCI-illiteracy phenomenon" [14].

Within BCI studies, neurophysiological and psychological variables (e.g., self-confidence and concentration) have been shown to significantly predict performance variation [15–17]. However, very little is known about the source of individual differences in the ability to volitionally regulate affective states. Therefore, the secondary aim of this project was to characterize individual variation in the ability to self-induce affective states using neurofeedback according to the subjects' unguided self-induction ability. This research has direct clinical relevance to informing our understanding of the neuroregulation capabilities of psychiatric patients to identify those most or least capable of guided affect regulation.

To explore our aims, we developed a novel task in which healthy adult participants utilized rtfMRI feedback to explicitly regulate their brain response and corollary affect processing states toward a goal of extreme pleasantness (i.e., positive valence). Attaining this brain-affect state triggered the presentation of an affectively congruent (positive valence) or incongruent (negative valence) image stimulus drawn from the International Affective Picture Set [18]

(IAPS). Between regulation trials participants passively viewed (without regulation) IAPS stimuli associated with either positive or negative valence. We then compared image stimulus-cued brain and affective responses arising from explicitly self-induced feedback-facilitated positive valence states versus random affective states (passive viewing) and tested the ability of self-induced positive valence states to bias the affect processing of subsequent image stimuli.

Our results reveal that self-induction of a positive affective state biases subsequent affect processing responses to image stimuli, suggesting a potential mechanism by which situational and attentional modification strategies work to reduce negatively biased affect processing states. We also found that individual differences in the intrinsic ability to self-induce affective arousal without guidance informed the attainment of self-induced positive valence in the presence of rtfMRI guidance, further supporting the established role of attentional deployment in explaining BCI performance.

## Methods

### Ethics statement

All participants provided written informed consent after receiving written and verbal descriptions of the study procedures, risks, and benefits. We performed all study procedures and analysis with approval and oversight of the Institutional Review Board at the University of Arkansas for Medical Sciences (UAMS) in accordance with the Declaration of Helsinki and relevant institutional guidelines and policies.

### Participants

We enrolled healthy adult participants (n = 40) having the following demographic characteristics: age [mean(s.d.)]: 38.8(13.3), range 20–65; sex: 22 (55%) female; race/ethnicity: 28 (70.%) self-reporting as White or Caucasian, 9 (22.5%) as Black or African-American, 1 (2.5%) as Asian, and 2 (5%) self-reporting as other; education [mean(s.d.)]: 16.8(2.2) years, range 12–23; WAIS-IV IQ [mean(s.d.)]: 102.5(15.3), range 73–129. All of the study's participants were right-handed (assessed via Edinburgh Handedness Inventory [19]) native-born United States citizens who were medically healthy and exhibited no current Axis I psychopathology, including mood disorders, as assessed by the SCID-IV clinical interview [4]. All participants reported no current use of psychotropic medications and produced a negative urine screen for drugs of abuse (cocaine, amphetamines, methamphetamines, marijuana, opiates, and benzodiazepines) immediately prior to both the clinical interview and MRI scan. When indicated, we corrected participants' vision to 20/20 using an MRI compatible lens system (MediGoggles™, Oxforshire, United Kingdom), and we excluded all participants endorsing color blindness.

### Experiment design

Following the provision of informed consent, subjects visited the Brain Imaging Research Center (BIRC) of the University of Arkansas for Medical Sciences on two separate days. On Study Day 1 a trained research assistant assessed all subjects for major medical and psychiatric disorders as well as administered instruments to collect data to be used as either secondary variables hypothesized to explain individual variance in affect regulation-related neural activity, covariates of no interest, or to assess inclusion/exclusion criteria. The participant returned to the BIRC for Study Day 2 within 30 days after Study Day 1 to complete the MRI acquisition. During this day, the participant received task training and completed the full MRI acquisition protocol, depicted in Fig 1.

**Fig 1. Study Day 2 experimental tasks: Order, number of repetitions, duration, and stimuli.** Tasks are colored by role. Gray depicts task training and application of psychophysiology recording apparatus. Blue depicts structural image acquisition. Orange depicts functional image acquisition. Identification and Modulation blocks of the fMRI acquisition summarize the relevant trial types used within that task (see Neuroimaging section for abbreviations). *Training of real-time multivariate pattern analysis predictive models was performed concurrently with the Resting State task of the fMRI acquisition.

**Training.** Each participant received a video-based overview of the experiment to be performed on that day as well as training on the study's task variations and trial types. The participant was offered the opportunity to use the restroom and then was moved to the MRI scanner room and fully outfitted with psychophysiological recording equipment.

**Neuroimaging.** For each subject we captured a registration scan and detailed T1-weighted structural image. We then acquired functional MRI data for three task variations: identification, resting state, and modulation. Identification (Id) task acquisition consisted of 2 x 9.4 min fMRI scans during which the participant was presented with 120 images drawn from the International Affective Picture System [18] (IAPS) to support one of two trial types (see Fig 2): 90 passive stimulus (PS) trials and 30 cued-recall (CR) trials. Identification task PS trials (abbreviated Id-PS) presented an image for 2 s (cue) succeeded by a fixation cross for a random intertrial interval (ITI) sampled uniformly from the range 2–6 s. Identification task cued-recall (Id-CR) trials were multi-part: a cue image was presented for 2 s followed by an active cue response step for 2 s (the word "FEEL" overlaying the image) followed by the word FEEL alone for 8 s, which signaled the participant to actively recall and re-experience the affective content of the cue image, followed by a 2–6 s ITI. During pre-scan training on the Id-CR task's recall condition, subjects were instructed to "Imagine the last picture you saw as best you can. Try to make yourself feel exactly how you felt when you saw this picture the first time. Hold that feeling the whole time you see the word FEEL." Within each scan, Id-PS and Id-CR trials were pseudo-randomly sequentially ordered to minimize correlations between the hemodynamic response function (HRF)-derived regressors of the tasks. This order was fixed for all subjects.

During resting state acquisition, we acquired 7.5 min of fMRI data in which the subject performed mind-wandering with eyes open while observing a fixation cross. During training, subjects were instructed to "Keep your eyes open, look at the cross in front of you, and let your brain think whatever it wants to." Concurrently with the resting state task, the real-time variant of the multivoxel pattern analysis (MVPA) prediction model (see below) was fit using data drawn from the Identification task fMRI data to define individual brain state representations of the affect processing goal.

Modulation (Mod) task acquisition consisted of 2 x 10.5 min fMRI scans during which the participant was presented with 60 IAPS images according to two trial types (see Fig 2): 40 passive stimulus (Mod-PS) trials, which were identically formatted to the Id-PS trials, and 20 feedback-triggered stimulus (Mod-FS) trials. Mod-FS trials used real-time fMRI feedback of the subject's decoded affective state to guide them in self-inducing affective brain states associated with their individualized representation of extreme positive valence. The computer system monitored the subject's decoded valence processing level at each acquisition volume of fMRI data and if that decoding met pre-defined criteria (i.e., the goal state, which we defined as hyperplane distance $\geq 0.8$ for 4 consecutive EPI volumes) then a positively (congruent) or

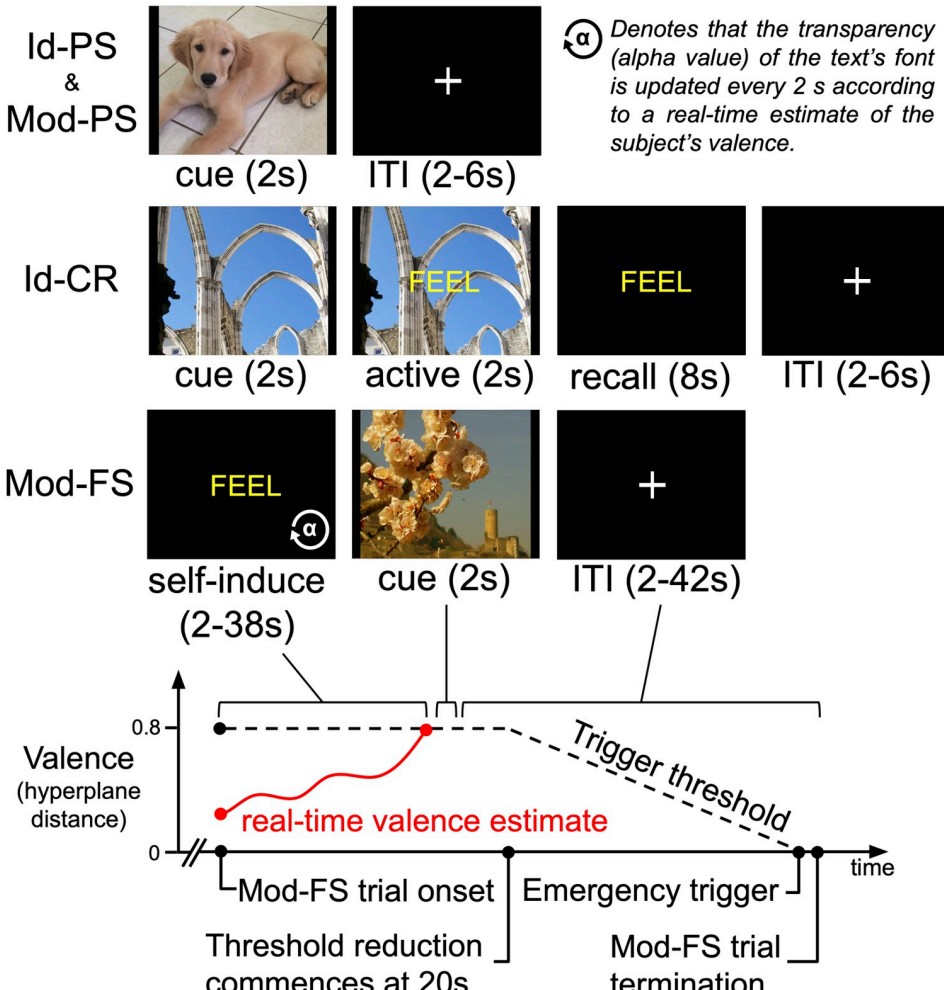

**Fig 2. Summary of experimental task trial designs.** (Id-PS): Identification task passive stimulus trials, which were identical to Modulation task passive stimulus (Mod-PS) trials. (Id-CR): Identification task cued-recall trials. (Mod-FS): Modulation task feedback-triggered stimulus trials. (Bottom): Depiction of a hypothetical Mod-FS trial for the experimental design. The dashed line represents the trigger threshold and bounds the hyperplane distance at which the cue stimulus will be triggered by the real-time valence estimate as a function of time. As depicted, this threshold decreases linearly to zero commencing at 20 s of feedback. This trial type is of fixed length; therefore, the ITI duration is a function of the time required to trigger the stimulus via feedback. If the real-time valence estimate does not surpass the trigger threshold prior to the threshold reaching zero then the stimulus is triggered by default, denoted "Emergency trigger", followed by the minimum ITI.

negatively (incongruent) valent image stimulus was triggered as the test stimulus. The brain state criteria representing the affect processing goal state were determined by the results of an initial pilot of the experiment to identify acquisition parameters that were challenging but consistently reachable. Within each scan, Mod-PS and Mod-FS trials were pseudo-randomly sequentially ordered to minimize correlations between the hemodynamic response function (HRF)-derived regressors of the tasks. This order was fixed for all subjects.

We provided real-time visual feedback during Mod-FS trials by manipulating the level of transparency of the word FEEL, which was the cue to volitionally regulate affect to an extreme positive valence. The transparency of the text was scaled to reflect real-time estimates of subject's represented valence processing with respect to the desired hyperplane distance threshold.

This was achieved by mapping MVPA prediction model hyperplane distances (see below) from their base range [-1.25,1.25] to the range of possible transparencies, $\alpha \in [0,1]$. Fully transparent text ($\alpha = 0$) appeared as a black screen and denoted poor affect regulation performance, i.e., highly negative valence. Fully opaque text ($\alpha = 1$) appeared bright yellow and denoted good performance. The transparency of the text was reset every 2 s (reflecting the momentary hyperplane distance prediction based upon each EPI volume, TR = 2000 ms). The transparency was adjusted (approximately 20 frames-per-second) to present smooth transitions toward the brain-affect goal state. The initial hyperplane distance threshold was fixed for 20 seconds. If the subject had not attained the threshold (i.e. triggered the test stimulus) by this time then the threshold was linearly and continuously lowered to 0 over the subsequent 18 s at which point the stimulus was automatically triggered even if the threshold had not been attained (Fig 2).

**Stimulus selection.** We sampled 180 IAPS images to use as affect processing induction stimuli. Identification task stimuli were sampled computationally using a previously published algorithm [20] that selects images such that the subspace of the valence-arousal plane for normative scores within the IAPS dataset is maximally spanned (see Fig 3). This property guarantees the most diverse range of valence and arousal properties for a fixed-sized stimulus set. We performed this full-range sampling process first for the 90 images used in Id-PS trials. The IAPS identifiers of these images were previously reported [21]. We then separately (but similarly) sampled an additional 30 images for use in Id-CR trials. The IAPS identifiers of these images were also previously reported [22]. Next, we constructed extreme polar subsets of positively and negatively valenced image stimuli by constructing thresholds of permissible valence and arousal scores. Valence (v) was constrained such that: v≥7 or v≤2.6. We then iteratively constrained the permissible arousal scores until we identified positively and negatively valent image subsets that did not exhibit a group mean difference in arousal, a, scores (found to be $4.6 < a < 6.8$) thereby controlling for arousal response as a stimulus subset variable. We then sampled 30 images each from these subsets and uniformly randomly assigned these images to Mod-PS trials (n = 40) and Mod-FS trials (n = 20), respectively. The outcomes of these sampling and assignment processes are presented in Fig 3. The specific IAPS identities of these images are reported in S1 Table.

## Data acquisition and processing

**MR image acquisition.** We acquired all imaging data using a Philips 3T Achieva X-series MRI scanner (Philips Healthcare, Eindhoven, The Netherlands) with a 32-channel head coil. We acquired anatomic images using an MPRAGE sequence (matrix = 256 x 256, 220 sagittal slices, TR/TE/FA = 8.0844/3.7010/8˚, final resolution = 0.94 x 0.94 x 1 mm$^3$). We acquired functional images using the following EPI sequence parameters: TR/TE/FA = 2000 ms/30 ms/90˚, FOV = 240 x 240 mm, matrix = 80 x 80, 37 oblique slices, ascending sequential slice acquisition, slice thickness = 2.5 mm with 0.5 mm gap, final resolution 3.0 x 3.0 x 3.0 mm$^3$.

**Real-time MRI preprocessing and multivariate pattern classification.** We implemented custom code that acquired each raw fMRI volume as it was written to disk by the MRI's computer system (post-reconstruction). Each volume underwent a preprocessing sequence using AFNI [23] in the following order: motion correction using rigid body alignment (corrected to the first volume of Identification task Run 1), detrending (re-meaned), spatial smoothing using a 8 mm FWHM Gaussian filter, and segmentation. To construct a multivariate pattern classifier to apply to the real-time data we partitioned the Id-PS stimuli into groups of positive and negative valence (according to the middle Likert normative score) and formed time-series by convolving the hemodynamic response function with the respective stimuli's onset times (scaling the HRF amplitude according to the absolute difference between the stimuli's

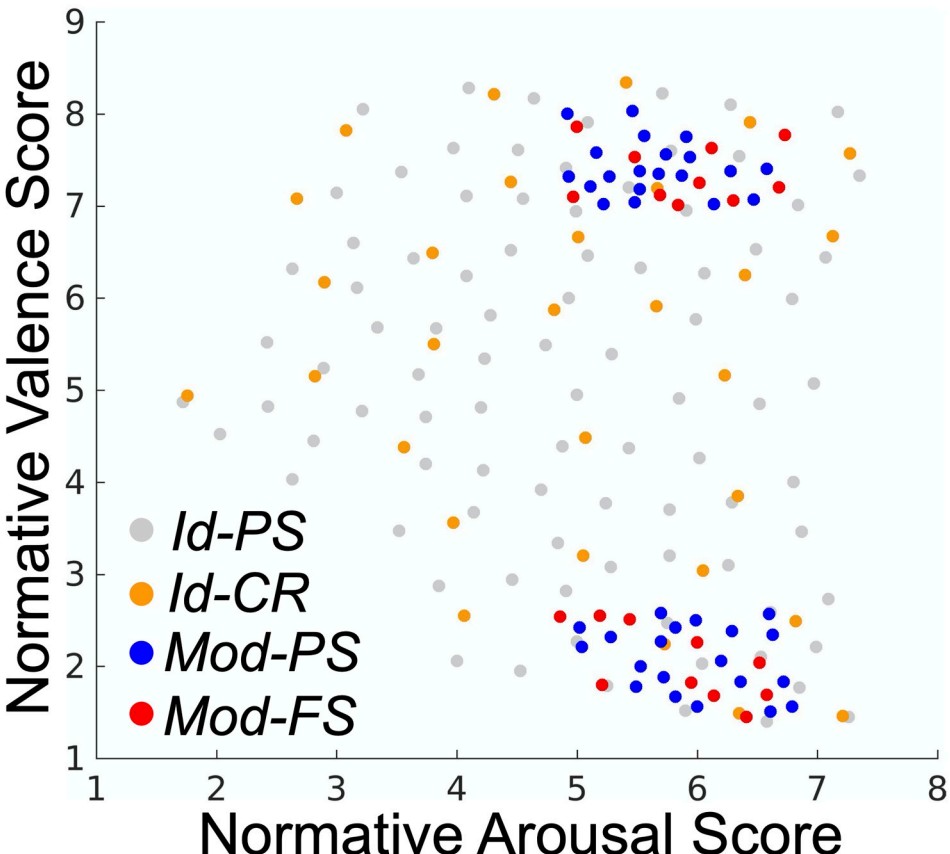

**Fig 3. Normative valence and arousal scores for stimuli selected for each of the four experimental trial types.**
Summary statistics for Identification task stimuli are as follows: Id-PS valence [mean (std. dev)] 5.04 (1.95); Id-PS arousal [mean (std. dev)] 4.95 (1.40); Id-CR valence [mean (std. dev)] 5.30 (1.95); Id-CR arousal [mean (std. dev)] 4.99 (1.51). There were no significant differences in affect properties between the Id-PS and Id-CR cue stimuli for either valence (p = .49; signed rank; α = .05; $h_0$: $\mu_1 = \mu_2$) or arousal (p = .86; rank-sum; α = .05; $h_0$: $\mu_1 = \mu_2$). Summary statistics for the Modulation task stimuli are as follows. Mod-PS (pos. valence cluster) valence [mean (std. dev)] 7.41 (.30); Mod-PS (neg. valence cluster) valence [mean (std. dev)] 2.08 (.36); Mod-FS (pos. valence cluster) valence [mean (std. dev)] 7.35 (0.32); Mod-FS (neg. valence cluster) valence [mean (std. dev)] 2.03 (0.41). Between the Mod-PS and Mod-FS stimuli in the positive valence cluster, there were no significant differences in valence (p = .60; rank-sum; α = .05; $h_0$: $\mu_1 = \mu_2$) nor arousal (p = .25; rank-sum; α = .05; $h_0$: $\mu_1 = \mu_2$). There were also no significant group differences in affect properties between the Mod-PS and Mod-FS stimuli in the negative valence cluster, either for valence (p = .74; rank-sum; α = .05; $h_0$: $\mu_1 = \mu_2$) or arousal (p = .54; rank-sum; α = .05; $h_0$: $\mu_1 = \mu_2$).

normative scores and the middle Likert score). We then thresholded these time-series to construct class labels {-1,+1} (as well as unlabeled) for each volume of the Identification task scans. We then trained a linear support vector machine [24] (SVM) to classify the valence property of each fMRI volume. Note, during the Modulation task the classification hyperplane output of the SVM was linearly detrended in real-time as follows. A hyperplane distance, h, was computed for each volume, i. For $h_i$, $i \geq 40$, the sequence of hyperplane distances $h_1, . . ., h_{i-1}$ was used to compute a linear trend (via the Matlab detrend function) which was subtracted from the hyperplane distance, $h_i$. In summary, the described system achieved real-time preprocessing and generated affect state predictions for each EPI volume acquired in the Modulation task of the experiment. Total processing time of each volume was less than the TR = 2 s parameter of the EPI sequence, allowing the real-time processing to maintain a consistent (reconstruction speed determined) latency throughout real-time acquisition.

**Post-hoc MRI preprocessing, multivariate pattern classification, and Platt-scaling.** We used fmriprep [25] (version 20.0.0) software to conduct anatomical and functional image pre-processing and spatial normalization to the MNI152 atlas (see S1 Methods for detailed documentation of this standarized image preprocessing pipeline). We then used fmriprep's motion parameter outputs to complete the preprocessing using AFNI, including regression of the mean time courses and temporal derivatives of the white matter (WM) and cerebrospinal fluid (CSF) masks as well as a 24-parameter motion model [26, 27], spatial smoothing (8 mm FWHM), detrending, temporal filtering (.0078 Hz high-pass), and scaling to percent signal change. For resting state functional images we took the additional step of global mean signal subtraction prior to smoothing.

We then conducted high-accuracy post-hoc multivoxel pattern analysis (MVPA), i.e., neural decoding, of affect processing. We first extracted beta-series [28] neural activation maps associated with Id-PS trials from fully preprocessed fMRI data recorded during Identification task runs 1 and 2 according to well-documented methods [20]. We indexed these maps according to their corresponding stimulus, x. Therefore, the maps, $\beta(x)$, were paired with their respective normative scores $\{\beta(x), v(x), a(x)\}$ to form training data for multivoxel pattern classification implemented via linear SVM. For classification training, valence and arousal scores were each converted into positive (+1) or negative (-1) class labels according to their relation to the middle Likert score. Classification hyperplane distances were then converted to probabilities (i.e., the probability of the positive class label) via Platt-scaling [29]. These probabilities served as the affective decodings of the subjects' brain states for further analysis.

**Affect processing state encodings.** In order to visualize affect processing brain states in neuroanatomical space, we performed a previously reported encoding transformation of our decoding models [21]. In short, we applied the Haufe-transform [30] to each subject's classification hyperplane and formed a map of group-level mean encoding values for each gray matter voxel. Separately, we generated 1,000 mean encoding permutations by applying the Haufe-transform to the classification hyperplanes fit to each subject's true beta-series and randomly permuted sets of the true affective labels. Those voxels exhibiting extreme group-level mean encoding values in comparison to the observed group-level mean permutation encoding values (2-sided test, $p < 0.05$) were kept for visualization of the brain state. We performed this encoding process separately for each dimension of affect processing (valence and arousal).

**Cued-recall, passive stimulus, and feedback-triggered stimulus modeling.** We also extracted beta-series for the cue and recall steps of the Id-CR trials, the cue step of the Mod-PS trials, and the cue step of the Mod-FS trials. We then used our fit SVM models to decode the valence and arousal properties of the subjects' brain states at these experiment steps. For the Mod-PS trials, we also constructed beta-series for the moment of trial onset as well as 2 s prior to the cue step of the Mod-FS trials–these allowed us to validate the triggers for affective stimulus test presentations as well as to measure (post-hoc) the relative change of affect processing achieved by feedback-facilitated self-induction of positive valence processing.

**Surrogate cued-recall task modeling.** Using previously reported methodology [31], we decoded the valence and arousal properties of each volume of Resting State fMRI data. We then uniformly randomly sampled 30 onset times for surrogate Id-CR trials and extracted the affect properties of the respective cue and recall steps of these surrogate trials to be used as within-subject controls during analysis of the actual Id-CR trials.

**Psychophysiology data acquisition and preprocessing.** All MRI acquisitions included concurrent psychophysiological recordings conducted using the BIOPAC MP150 Data Acquisition System and AcqKnowledge software combined with the EDA100C-MRI module (skin conductance), TSD200-MRI pulse plethysmogram (heart rate), TSD221-MRI belt (respiration), and EMG100C-MRI module (facial electromyography). In line with prior work [32, 33],

we measured arousal independently based on skin conductance response (SCR) and valence based on facial electromyography (fEMG) response, specifically activity in the corrugator supercilli muscle (cEMG), which was shown in prior work to capture the full affective valence range of our affect processing induction design [22]. This work did not model the heart and respiratory rate data. We have extensively reported on our SCR electrode placement and pre-processing methods [21], and we recently reported our cEMG placement and preprocessing methods [22].

## Results

### Psychophysiological response validation of affect processing induction via image stimuli

We first verified the ability of the Identification task passive stimulus (Id-PS) trials to induce corollary psychophysiological responses [34] associated with affect processing in order to validate the inputs used to train our neural decoding models. We modeled the normative scores of the cue stimuli of Id-PS trials using psychophysiological response measures within a General Linear Mixed-Effects Model (GLMM) framework, respectively, for valence and arousal properties. Normative hedonic valence scores of the stimuli were modeled according to facial electromyographic responses in the corrugator supercilli as the fixed effects. Normative autonomic arousal scores to the cue stimuli were modeled according to skin conductance responses as the fixed effects. In both models, we controlled for age and sex effects. Slope and intercept random-effects were modeled subject-wise. Both validation models detected significant stimulus-related induction of the anticipated physiological responses. Moreover, our cEMG-derived model of hedonic valence ($\beta = .11$; $p = .001$; t-test; $\alpha = .05$; $h_0$: $\beta = 0$) was selective for the valence property of affect–a cEMG-derived model of autonomic arousal was not significant ($p = .75$; t-test; $\alpha = .05$; $h_0$: $\beta = 0$). Similarly, our SCR-derived model was selective for the autonomic arousal property of affect ($\beta = .07$; $p = .004$; t-test; $\alpha = .05$; $h_0$: $\beta = 0$)–applied to hedonic valence the SCR associations were not significant ($\beta = .02$; $p = .61$; t-test; $\alpha = .05$; $h_0$: $\beta = 0$). These results are consistent with the prior association of cEMG and SCR with the processing of the specific affect properties of valence and arousal, respectively, and support the induction of affect processing during the Id-PS trials.

### Affect processing measurement

We next demonstrated that our prediction models accurately decoded affect processing within neural activation patterns associated with Id-PS trials, reproducing the results of earlier work using similar modeling methodology [20]. Our tabulated prediction accuracy (averaged over 39 subjects completing the experiment) over the full stimulus set (see Table 1) was highly significant for both valence ($p < .001$; signed rank; $\alpha = .05$; $h_0$: $\mu = .5$) and arousal ($p < .001$; signed rank; $\alpha = .05$; $h_0$: $\mu = .5$). We also observed prediction performance comparable to the best known demonstrations of neural decoding of affect processing across the valence and arousal dimensions [20, 35] when our measurements were restricted to those image stimuli

**Table 1. Multivariate neural decoding performance.**

| | Valence | Arousal |
|---|:---:|:---:|
| | Grp. Avg. Acc. (95% CI) | Grp. Avg. Acc. (95% CI) |
| **Full Stimulus Set** | .55 (.53,.57) | .61 (.59,.63) |
| **Reliable Stimulus Set** | .79 (.76,.82) | .75 (.72,.79) |

exhibiting reliable brain state activations, i.e., the reliable stimulus set (see Table 1), which were determined according to previously published methods [20] that detect the degree to which brain states induced by these stimuli cluster between subjects (see S1 Methods). Indeed, using the reliable stimulus set to measure performance, we found that 34 of 39 subjects (87.2%) exhibited significant within-subject classification of affective valence and arousal stimuli, respectively ($\alpha$ = .05; binomial distribution, $h_0$: p[+] = .5). These results support the validity of our neural decoding models as brain representations of affective valence and arousal.

## Validation of affect decoding using novel stimuli

Prior to applying our decoding models to novel task domains, we first tested whether these models (originally fit to Id-PS features and labels) generalized to novel image stimuli. To perform this independent test we modeled, via GLMM, the normative affect scores of cue stimuli in Id-CR and Mod-PS trials. However, each test was unique.

First, we modeled Id-CR task cue stimuli's normative scores as a function of decoded affect (separately for valence and arousal) controlling for the age and sex of the subjects and modeling random effects of affect decoding subject-wise. In Id-CR trials we found that neurally decoded valence was significantly positively associated with the valence normative score ($\beta$ = .30; p < .001; t-test; $\alpha$ = .05; $h_0$: $\beta$ = 0). Similarly, we found for Id-CR trials that neurally decoded arousal was significantly associated with the arousal normative score ($\beta$ = .17; p = .001; t-test; $\alpha$ = .05; $h_0$: $\beta$ = 0). Age and sex effects in both cases were not significant and random effects did not significantly improve the model's explained variance, which was very small for both valence ($R^2_{adj}$ = .02) and arousal ($R^2_{adj}$ = .01), respectively.

Next, we modeled the Mod-PS task stimuli's normative scores as a function of decoded affect (separately for valence and arousal normative scores). However, in this case we controlled for age and sex effects as well as the decoding of the complementary affective property in order to control for the bias of the sampling of the stimuli in this task (see Fig 3). In Mod-PS trials we found that decoded valence was significantly positively associated with the stimuli's normative valence scores ($\beta$ = .62; p < .001; t-test; $\alpha$ = .05; $h_0$: $\beta$ = 0). However, decoded arousal was significantly negatively associated with normative valence scores ($\beta$ = -.22; p = .016; t-test; $\alpha$ = .05; $h_0$: $\beta$ = 0). Age and sex effects were not significant but random effects did significantly improve the model's explained variance ($R^2_{adj}$ = .045). In contrast, we found no significant associations between decoded arousal and the stimuli's normative arousal scores, which confirmed that the restriction of our sampling of the Mod-PS and Mod-FS stimuli to a narrow range of normative arousal (see Fig 3) was essential as a control for this confounding variable.

## Validating the rigor and reproducibility of affective brain states

In a final validation step, we sought to provide additional qualitative and quantitative evidence for the rigor and reproducibility of the affective brain states that we experimentally manipulated in this study. We computed the group-level encodings of both the arousal and valence brain states that survive permutation testing, which we present in Fig 4. Encodings of affect processing largely overlap with earlier multivariate [21] and univariate meta-analyses [36, 37] of the neural encoding of core affect processing. We took the additional step of directly comparing these encodings to affect processing encodings that were computed for past studies that incorporated similar affect induction stimuli and used similar fMRI analysis pipelines but that were derived from separate sets of research subjects (see S1 Methods). Notably, these past studies found that affect processing predictions using the machine learning models underlying

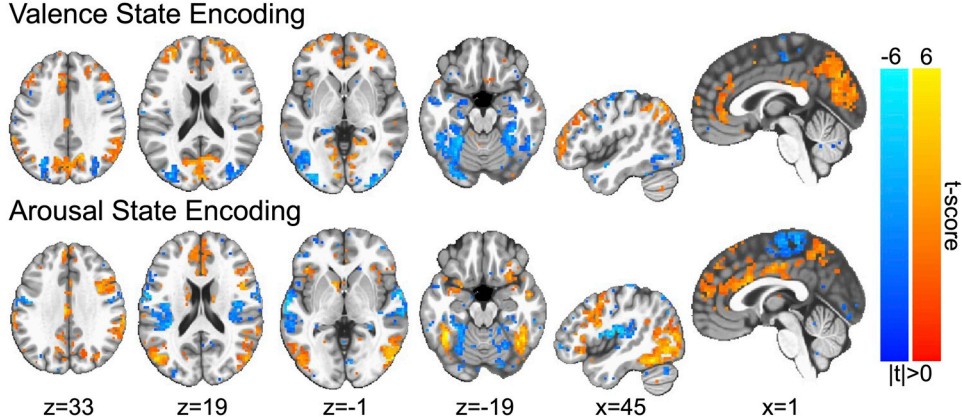

**Fig 4. Group-level encodings of affective state processing.** Color gradations indicate the group-level t-scores of the encoding parameters (red indicating positive valence or high arousal, blue indicating negative valence or low arousal). T-scores are presented only for those voxels in which encoding parameters survived global permutation testing (p < .05). Image slices are presented in MNI coordinate space and neurological convention. Maximum voxel intensity is |t| = 6.0, i.e., color saturates for t-scores with absolute values falling above this value.

these encodings were significantly more correlated to the normative scores of the induction stimuli than predictive measures derived from psychophysiological responses across the independent dimensions of affective valence (measured via heart-rate deceleration [38]) and arousal (measured via skin conductance response [21]). Indeed, we found that the neural encodings computed for this study shared 36.5% of the variance across prior whole-brain gray-matter voxel-wise encodings of valence as well as 31.1% of the variance across prior whole-brain voxel-wise encodings of arousal (see S1 Fig). Of note, the variance shared between these encodings rose to 87.0% and 85.6%, respectively for valence and arousal, when we restricted the comparison to only those voxels that survived global permutation testing (i.e., the voxels presented in Fig 4).

## Real-time stimulus triggering

We next validated that our real-time feedback and brain-affect state triggering process functioned as designed. To test this we extracted the feedback signal calculated at the moment of stimulus trigger (including emergency triggering). The median feedback at the moment of trigger was $\mu = .93$ (p < .001; signed rank; $\alpha = .05$; $h_0$: $\mu = 0$). Nearly three-quarters (see Fig 5) of all trials triggered at or above the design threshold.

## Real-time fMRI-guided self-induction of positive valence states

We next demonstrated that our primary experimental manipulation, volitionally-induced positive valence, was truly achieved at the moment of stimulus triggering. As a reminder, the Mod-FS trials were triggered using lower quality real-time affect decoding models. Here we applied post-hoc high-accuracy models to decode affect processing within the fMRI volume immediately prior to the stimulus trigger as a best possible measure of the experimental condition. However, a confounding factor of this measure is within-subject valence decoding accuracy, which we found to significantly positively associate with the magnitude of decoded valence at the moment of real-time stimulus triggering (see S2 Fig) and, therefore, could potentially act as a confound of the experimental manipulation Therefore, to test this measure we bootstrapped random variants of the trigger predictions (randomly sampling within each

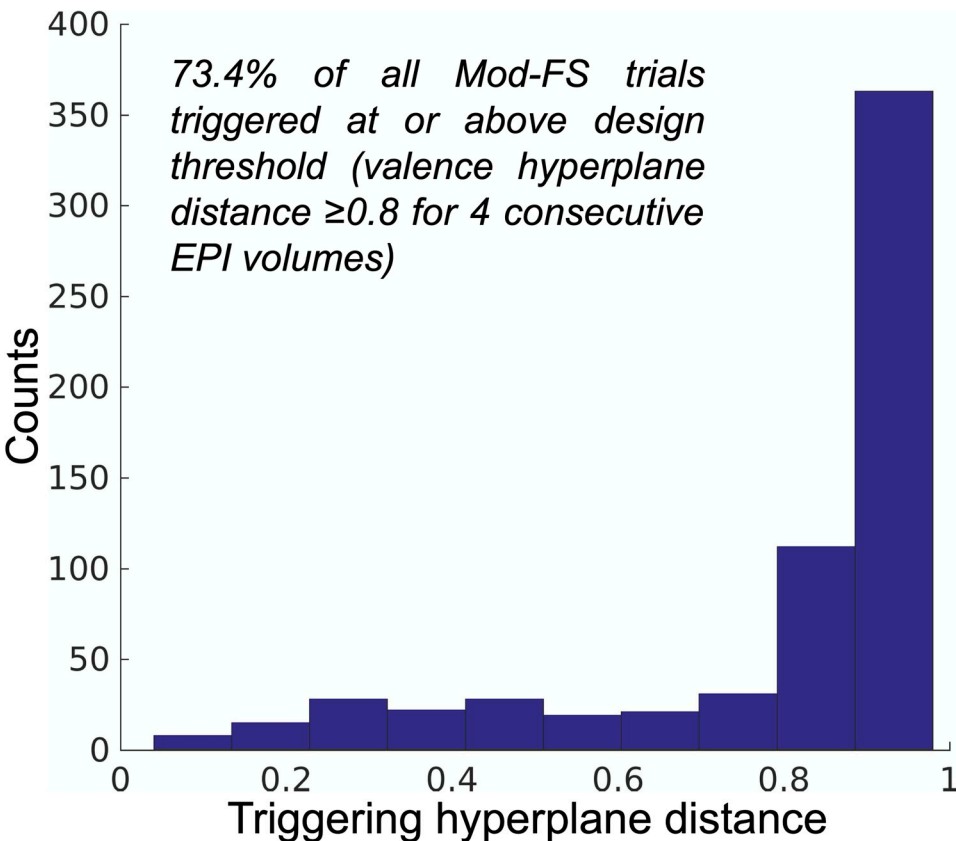

**Fig 5. Distribution of average feedback scores at the moment of FT-PO trial stimulus trigger.**

subject before pooling predictions to incorporate random effects) for only those subjects exhibiting within-subject significant decodings of valence processing. From these neural decodings, we found that the mean predicted valence was significantly elevated ($\mu = .522$; $p < .02$; 1-sided bootstrap [n = 10000]; $h_0$: $\mu < .5$) at the time of triggering of the test stimuli. Independently, we confirmed that volitionally-induced positive valence states corresponded with significant changes to independent psychophysiological response measures across all subjects, including those that did not exhibit within-subject significant decodings of valence processing. In concordance with our observations of psychophysiological responses induced by extrinsic image stimuli, self-induction of positive valence induced a weak but significant positive cEMG response ($\beta = .003$; $p < .01$; t-test; $\alpha = .05$; $h_0$: $\beta = 0$) as well as a significant reduction in SCR ($\beta = -.018$; $p < .001$; t-test; $\alpha = .05$; $h_0$: $\beta = 0$).

### Effect of positive valence self-induction on affect processing of subsequent stimuli

We next tested the study's primary hypothesis–that self-induced states of positive valence bias the affect processing of subsequent image stimuli. Using a GLMM, we tested decoded valence processing of these stimuli as a function of trial type, Mod-FS (i.e., self-induced) or Mod-PS (passive), while controlling for the image stimuli's associated normative valence and decoded arousal properties, the subject's age and sex, as well as within-subject valence decoding accuracy. To control for potential confounding effects of the slow temporal evolution of the HRF, we also included the decoded valence of the volume immediately preceding the image stimulus

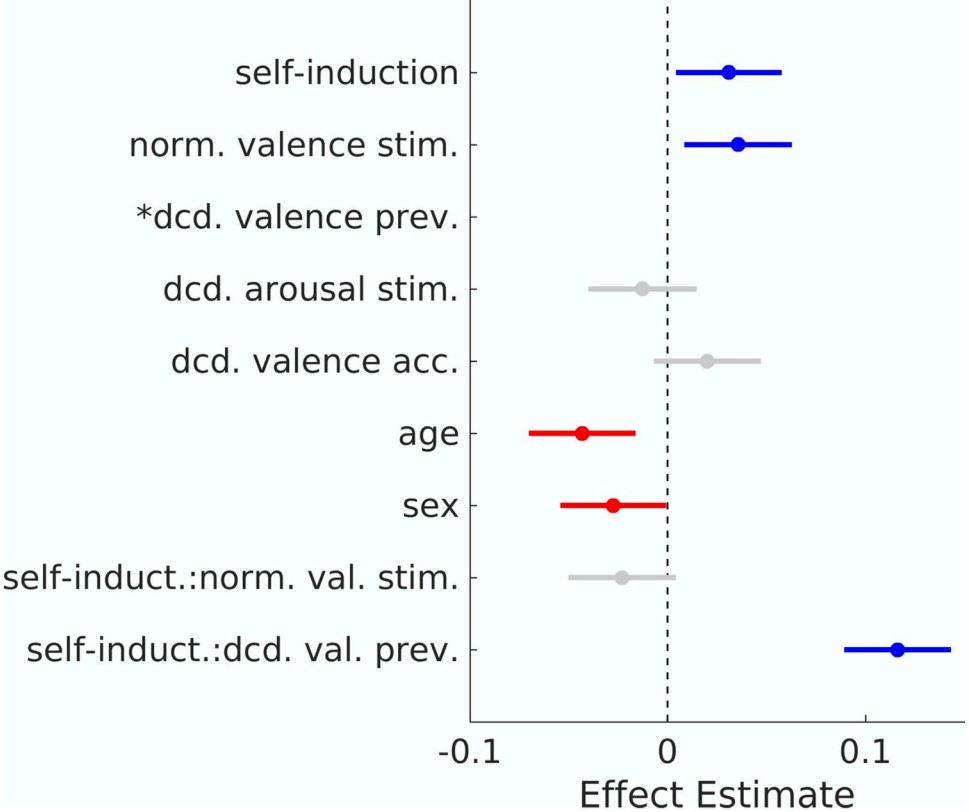

**Fig 6. Effects of volitional self-induction of positive valence on affect processing bias of subsequent image stimuli.**
The figure graphically depicts the effect sizes estimated for the primary experimental manipulation, i.e. the self-induction trial type (feedback-triggered stimulus, Mod-FS, versus passive stimulus, Mod-PS), denoted self-induction, on the decoded valence processing of the subsequent image stimulus while controlling for the effects of the normative valence score of the stimulus, the decoded valence processing of the previous fMRI volume, normative arousal score of the image stimulus as well as subjects' age and sex and the two-way interactions between self-induction trial type and the normative valence score of the stimulus as well as the decoded valence of the previous fMRI volume. Statistically significant effects are colored blue (positive effects) or red (negative effects). Non-significant effects are colored gray.
*The effect size of the decoded valence processing ($\beta$ = .802) of the previous fMRI volume was omitted from the figure to elevate the contrast between the smaller effect sizes.

(i.e., the trigger fMRI volume in Mod-FS trials or the previous fMRI volume in Mod-PS trials) as a fixed effect. Finally, we included two-way interactions between the trial type and both the normative valence score of the image stimulus and the decoded valence of the preceding fMRI volume. We modeled random intercept effects subject-wise.

We found that the volitional self-induction of positive valence prior to an affective stimulus significantly positively biased the induced valence processing of the subsequent image stimulus ($\beta$ = .033; p = .017; t-test; $\alpha$ = .05; $h_0$: $\beta$ = 0) compared with passive viewing. As would also be expected, normative valence score of the stimulus was significantly positively associated with valence processing ($\beta$ = .031; p = .027; t-test; $\alpha$ = .05; $h_0$: $\beta$ = 0) as was the decoded valence of the previous volume ($\beta$ = .803; p < .001; t-test; $\alpha$ = .05; $h_0$: $\beta$ = 0). For clarity, the magnitude of the effects are depicted graphically in Fig 6. Both sex ($\beta$ = -.028; p = .039; t-test; $\alpha$ = .05; $h_0$: $\beta$ = 0) and age ($\beta$ = -.043; p = .002; t-test; $\alpha$ = .05; $h_0$: $\beta$ = 0) were significantly negatively associated with valence processing of the subsequent image stimulus. Finally, the stimuli's normative arousal scores were found not to be a significant predictor of decoded valence processing ($\beta$ = -.028; p = .051; t-test; $\alpha$ = .05; $h_0$: $\beta$ = 0) nor was within-subject valence decoding model

accuracy ($\beta$ = .020; p = .15; t-test; $\alpha$ = .05; $h_0$: $\beta$ = 0). We did observe a significant interaction between self-induction trials and the decoded valence of the preceding fMRI volume ($\beta$ = .117; p < .001; t-test; $\alpha$ = .05; $h_0$: $\beta$ = 0); however, the interaction between trial type and the normative valence score of the image stimulus was not significant ($\beta$ = -.019; p = .197; t-test; $\alpha$ = .05; $h_0$: $\beta$ = 0). Overall model performance was $R^2_{adj}$ = .682 and random effects did not significantly impact the model's explained variance (p < .05; likelihood ratio test; $h_0$: observed responses generated by fixed-effects only).

As an independent exploration of our experimental manipulation, we repeated the study's primary hypothesis test using psychophysiological response measures of affect processing, respectively SCR and cEMG, as the measures of interest in GLMM models while controlling for similar fixed, interaction, and random effects as were used in the neuroimaging analysis. Using these models, we found that volitionally-induced positive valence did not significantly bias cEMG responses to image-based affect induction. However, SCR response measures to subsequent image stimuli were positively associated with both the primary experimental manipulation ($\beta$ = .141; p < .001; t-test; $\alpha$ = .05; $h_0$: $\beta$ = 0) and normative valence score of the subsequent image stimulus ($\beta$ = .042; p = .046; t-test; $\alpha$ = .05; $h_0$: $\beta$ = 0). No other effects were significant. Overall, the model's explained variance was $R^2_{adj}$ = .019 and random effects did not significantly impact the model's performance.

## Measurement of unguided explicit affect regulation

We next sought to confirm affect self-induction via unguided explicit (i.e. effortful) affect regulation within the Id-CR trials. We first decoded the valence and arousal responses from acquired fMRI data for both the cue and recall steps of the Id-CR trials. We then tested for group effects of explicit affect regulation toward a known goal by modeling via GLMM, separately for valence and arousal, the neurally decoded affect processing of the four recall steps of the Id-CR trials (4 volumes, 2 seconds each) as a function of the neurally decoded affect processing associated with the cue stimuli (i.e. the affect regulation goal) as well as the control duration and the age and sex of the subject (see Fig 7). We found that the subjects significantly regulated brain representations of valence processing ($\beta$ = .33; p < .001; t-test; $\alpha$ = .05; $h_0$: $\beta$ =

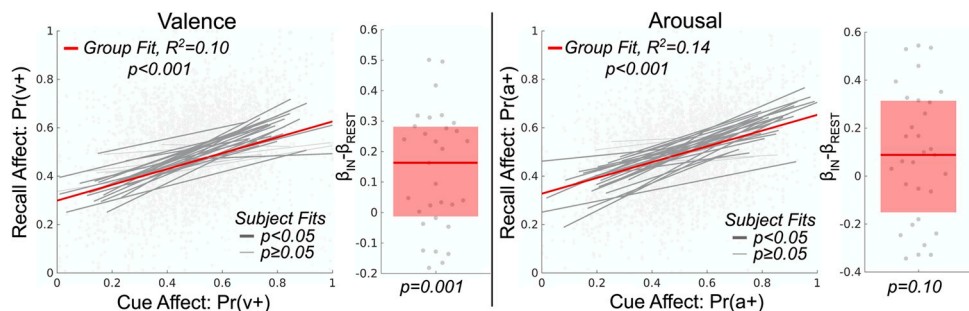

**Fig 7. Estimation and validation of explicit intrinsic affect regulation effects within the cued-recall task.** The figure depicts the effect size of cue affect processing in explaining affect processing occurring during recall (controlling for time lag in the 4 repeated measures of recall per each measure of cue). Here affect processing measurements are Platt-scaled hyperplane distance predictions, Pr($\cdot$), of our fitted support vector machine models. Valence and arousal dimensions of affect are predicted by separate models. The figure's scatterplots depict the group-level effects computed using linear mixed-effects models which model random effects subject-wise. Bold red lines depict group-level fixed-effects of the cue affect. Bold gray lines depict significant subject-level effects whereas light gray lines depict subject-level effects that were not significant. The figure's boxplots depict the group-level difference between each subject's affect regulation measured during the cued-recall trials in comparison to surrogate affect regulation constructed from the resting state task. The bold red line depicts the group median difference in effect size between task and surrogate. The red box depicts the 25-75th percentiles of effect size difference.

0). Random effects significantly improved the model's effect-size (p < .05; likelihood ratio test; $h_0$: observed responses generated by fixed-effects only) and cued-recall affect regulation effects were significantly greater than that of surrogate (control) effects (p = .001; signed rank; α = .05; $h_0$: $\beta_{IN}$-$\beta_{RST}$ = 0). The fixed-effect of control duration was also significant (β = .01; p < .001; t-test; α = .05; $h_0$: β = 0) and the overall model prediction performance was good ($R^2_{adj}$ = .10). Further, we found that subjects significantly regulated the neural correlates of arousal responses and that random effects significantly improved effect-size (β = .33; p < .05; likelihood ratio test; $h_0$: observed responses generated by fixed-effects only); however, these cued-recall affect regulation effects were not significantly greater than that of surrogate effects (p = .10; signed rank; α = .05; $h_0$: $\beta_{IN}$- $\beta_{RST}$ = 0).

## Unguided explicit affect regulation performance as a predictor of rtfMRI-guided self-induction

Finally, we tested whether unguided explicit affect regulation performance explained the level of rtfMRI-guided self-induced valence responses (measured immediately prior to presentation of the Mod-FS cue image). We modeled the neurally decoded valence of the final volume of the self-induce step of Mod-FS trials (see Fig 2) as a function of the individual subjects' explicit affect regulation performance parameters (slope and intercept, respectively, for the valence and arousal properties of affect processing–see Fig 7) controlling for the subjects' age, sex and valence decoding accuracy. We included all 2-way interactions between the slope and intercept fixed effects in this model to control for potential trade-offs that the subjects may be making during explicit regulation, e.g., focusing on only one affective property. We also included 2-way interactions of valence slope and intercept with age, sex, and valence decoding accuracy fixed effects. We found that self-induced arousal slope, i.e., the ability of the subject to accurately match the relative affective arousal of the goal, was significantly associated with rtfMRI-guided self-induced valence responses (β = .850; p = .004; t-test; α = .05; $h_0$: β = 0). However, the total explained variance by this model was very low ($R^2_{adj}$ = .002).

## Discussion

This work made two novel contributions to our current and future understanding of the mechanisms of emotion processing and regulation. First, we found significant support for the utility of self-induced positively valent affect processing as a mechanism for positively biasing the subsequent valence processing of environmental stimuli. This finding mechanistically supports the common notion of "positive thinking" and provides insight into how and why attentional re-deployment strategies, e.g. positive distraction, may benefit those suffering from deficits of emotion regulation and dispositional negatively biased affect. Second, we demonstrated a novel application of real-time brain state decoding in which we guided subjects' explicit emotion regulation toward a pre-defined affective goal state (positive valence) and then triggered experimental stimuli when the subjects' affective states fell within designed criteria representing that goal state. This new technology, while still in its infancy, may provide scientists with a much needed tool for exploration of intrinsic emotion processing mechanisms and their relationships with other cognitive processes and environmental factors.

The validity of our findings, as well as the efficacy of the proposed real-time affect processing decoding technology, are supported by independently measured psychophysiological responses at each stage of the experimental manipulation. Significant psychophysiological response correlates of affect processing (measured as SCR and cEMG) were observed during image-based induction of affect processing brain states (on which the neural decoding models were trained) as well as during volitional self-induction of positive valence processing. Moreover, skin conductance

responses to image-based affect processing induction differed significantly between the conditions of our primary experimental manipulation: feedback-triggered (Mod-FS) versus passively triggered (Mod-PS) image stimuli. As these effects were computed from independent processes operating on unique time-scales from those of the HRF, these findings suggest that our analyses are robust to the time-scales by which canonical response functions evolve and confers support for the primary neural decoding effects that we report in Fig 6.

A secondary goal of this work was to explain individual differences observed in real-time fMRI guided explicit emotion regulation toward a defined goal. Explicit affect regulation can be achieved volitionally, without the use of neurofeedback technology. Therefore, our use of real-time fMRI-based affective decodings to guide (or focus) this innate process enabled us to test (using unguided explicit affect regulation ability as a baseline) the association between innate affect regulation performance and the performance achievable using our real-time fMRI feedback approach. We observed a small but significant relationship between the ability to match one's arousal to a pre-defined target level and the ability to self-induce positive valence via rtfMRI-guidance. These findings suggest that subjects with greater control over their state of arousal exhibit improved ability to incorporate real-time feedback. Given the well-established link between arousal and attention [39, 40], these findings may in turn reflect improved deployment of attention, either self-directed or with respect to the feedback signal, in subjects exhibiting superior rtfMRI-guided self-induced valence, which agrees with earlier work in identifying psychological predictors of BCI performance [16, 41].

Our application of neural decodings (derived from normative affective scores of IAPS image stimuli) as markers of affect processing has well-known limitations, which we have noted in earlier reports[20, 21, 38]. Indeed, our validation process detected a significant negative effect of decoded arousal associated with decoded valence, suggesting that our cohort of subjects perceived the affective content of Mod-PS image stimuli differently than that which was captured by the IAPS normative scores. However, the nature of our investigation–real-time moment-to-moment affect processing, regulation, and stimulus-triggering–did not, unfortunately, permit the use of subject self-report measures of affect, thereby precluding a full concordance of our findings across cognitive, physiological, and behavioral domains. We also acknowledge technical limitations in our real-time fMRI approach. Despite significant findings of an overall effect, we believe that our implementation was suboptimal due both to response-measurement latency as well as perhaps insufficient optimization of parameters within our real-time pipeline. A limitation of real-time approaches is that parametric choices in the processing pipeline (e.g., trigger threshold) interact with experimental outcomes; therefore, it is difficult to use batch-wise optimization to inform the design criteria *a priori*. Moreover, our small study sample did not permit sufficient piloting of parameters prior to selecting the processing design and testing. Further, our analysis included all rtfMRI-guided self-induction trials, even those that required emergency triggering due to a failure to meet the design criteria of the goal state. This was intentional in order to put forth the most conservative, and therefore reproducible, estimate of the valence self-induction effect sizes possible using this new technological approach. Therefore, we believe the performance of the system, and its effect sizes, are understated, which suggests the potential to further refine this technology for larger-scaled deployment of brain-state driven experiment designs to test interactions between internal cognitions and external stimuli.

## Conclusion

We combined established neural decoding methods with real-time fMRI to construct a dynamic experimental design in which the brain representation of a subject's self-induced

positive affect state triggered the randomized presentation of affectively congruent or incongruent image stimuli. We first validated the experiment's ability to induce affect processing with independent measures of psychophysiology as well as the decoding models' ability to predict affect processing in novel task domains. We then demonstrated that self-induced positive affective states positively bias the affect processing of subsequent image stimuli and thereby furnish a mechanism by which positive thinking influences how we perceive our environment.

## Supporting information

**S1 Methods. Supplemental materials and methods.**
(DOCX)

**S1 Fig. Comparing similarity of the derived neural encodings of affective image stimuli across multiple studies. (Top Row):** Inter-study comparison that includes the encoding values of all joint GM voxels shown for (A) affective valence processing and (B) affective arousal processing. **(Bottom Row):** Inter-study comparison that include encoding values for only those joint GM voxels that survive global permutation significance testing ($p < .05$) for (C) affective valence processing and (D) affective arousal processing. Voxel-wise relationships are depicted as gray circles. The regression fit of the voxel-wise relationships is represented by the bold red line in each subplot. Total surviving joint voxels for each comparison are provided in the top left of each subplot. Inter-study shared variance is provided in the bottom right of each subplot. P-values refer to the significance of the regression fit's linear coefficient (t-test, $\alpha = 0.05$).
(TIF)

**S2 Fig. Effect of post-hoc decoding model accuracy on self-induction.** (Left) The magnitude of real-time self-induced positive affect processing (Mod-FS trials) according to the method of iteratively reweighted least squares versus. The measure of interest is Platt-scaled decoded valence observed at the moment of stimulus triggering. The fixed effect is the valence decoding model accuracy (measured according to the Full stimulus Set). Post-hoc decoding accuracy, which potentially reflects real-time decoding and, therefore, self-induction performance, was found to have a small ($R^2 = 0.009$) but significant positive effect on the decoded valence at the moment of real-time stimulus triggering ($\beta = .23$; $p = 0.014$; t-test; $\alpha = .05$; $h_0$: $\beta = 0$). (Right) The effect of post-hoc decoding model accuracy on the magnitude of random affect processing occurring in the fMRI volume acquired immediately prior to passive image stimulation (Mod-PS trials). No significant effects were observed ($\beta = .09$; $p = 0.099$; t-test; $\alpha = .05$; $h_0$: $\beta = 0$).
(TIF)

**S1 Table. International Affective Picture Set Image Identification Numbers.** IAPS identification numbers listed separately for Mod-PS and Mod-FS trial types.
(DOCX)

**S1 Dataset. Figure datasets.** Raw data files accompanying each figure.
(ZIP)

**S2 Dataset. Regression models.** Matlab formatted regression model files.
(ZIP)

## Acknowledgments

The authors thank Kevin Fialkowski and Ivan Messias for their help in curating project data and Maegan Calvert for her thoughtful comments on the manuscript. The authors would also

like to thank Kayla A. Wilson, Anthony A. Privratsky, Bradford S. Martins, Jennifer Payne, Emily Hahn, Natalie Morris, Nathan Jones, and Laura Spell for their assistance in recruiting and assessing research subjects and acquiring subject data as well as Stephen LaConte and Jonathan Lisinski for their assistance in developing our real-time fMRI capability. Finally, the authors thank Favrin Smith for her efforts in gaining the study's IRB protocol approval and maintaining human subject research compliance throughout the study's duration.

## Author Contributions

**Conceptualization:** Keith A. Bush.

**Data curation:** Keith A. Bush.

**Formal analysis:** Keith A. Bush.

**Funding acquisition:** Keith A. Bush.

**Investigation:** Keith A. Bush.

**Methodology:** Keith A. Bush.

**Resources:** Clinton D. Kilts.

**Software:** Keith A. Bush.

**Writing – original draft:** Keith A. Bush, Clinton D. Kilts.

**Writing – review & editing:** Keith A. Bush, Clinton D. Kilts.

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
