## [Decision Letter · Decision Letter 0]

23 Nov 2021

PONE-D-21-19149

A causal test of affect processing bias in response to affect regulation

PLOS ONE

Dear Dr. Bush,

Thank you for submitting your manuscript to PLOS ONE. After careful consideration, we feel that it has merit but does not fully meet PLOS ONE’s publication criteria as it currently stands. Therefore, we invite you to submit a revised version of the manuscript that addresses the points raised during the review process.

The critiques from the 1st reviewer appear to be highly addressable whereas the issues raised in the 2nd critique are potentially more serious red flags.

We look forward to receiving your revised manuscript.

Kind regards,

Desmond J. Oathes

Academic Editor

PLOS ONE

Journal Requirements:

2. We note that Figure 2 in your submission contain copyrighted images. All PLOS content is published under the Creative Commons Attribution License (CC BY 4.0), which means that the manuscript, images, and Supporting Information files will be freely available online, and any third party is permitted to access, download, copy, distribute, and use these materials in any way, even commercially, with proper attribution. For more information, see our copyright guidelines: http://journals.plos.org/plosone/s/licenses-and-copyright.

Reviewers' comments:

Reviewer's Responses to Questions

**Comments to the Author**

1. Is the manuscript technically sound, and do the data support the conclusions?

Reviewer #1: Partly

Reviewer #2: Yes

2. Has the statistical analysis been performed appropriately and rigorously? 

Reviewer #1: Yes

Reviewer #2: Yes

3. Have the authors made all data underlying the findings in their manuscript fully available?

Reviewer #1: Yes

Reviewer #2: Yes

4. Is the manuscript presented in an intelligible fashion and written in standard English?

Reviewer #1: Yes

Reviewer #2: No

5. Review Comments to the Author

Reviewer #1: This is a technically impressive investigation of the potential for fMRI to decode affective states from pictorial images and to utilize this information to track and trigger subsequent presentation of affective stimuli.

I have several major comments and then some additional areas for clarification:

1). One conceptual issue I note as problematic in the presentation and framing of these results is that the design and execution of this task somehow replicates or captures key ingredients of CBT. With some notable exceptions (e.g. reminiscence therapy), CBT does not require or engage the participant in deliberate up-regulation of positive emotion via internally-generated strategies, but it instead focuses on realistic appraisal of thoughts and situations with the typically focus on removing or altering distorted interpretations that tend to support and generate negative affect. This is the typical form of cognitive reappraisal as practiced in numerous CBT treatments, and it is not about “positive thinking” but “realistic thinking.” The approach utilized here is qualitatively different, and I think this needs to be better explained in the Introduction and Discussion.

2). The use of the word “causal” in the title and throughout the manuscript is too provocative and controversial a term and does not accurately reflect the approach utilized in this study. There are numerous arguments about how causality can be inferred and what constitutes causal inference, but I believe the authors are going too far with the use of this term given the experimental design utilized here. This term should be removed in favor of more accurate descriptions such as “experimental manipulation.”

3). Regarding the claim that self-induced positive affective states positively bias the affect processing of subsequent image stimuli, I think there should be more consideration and nuance with this interpretation. First, the coarse time resolution of the fMRI protocol utilized here (and the slow hemodynamic response function more generally) doesn’t allow for a clean separation of carryover effects form the positive affect up-regulation period to the processing of the triggered stimulus cue, and it’s not clear how these two periods interact in the resultant brain activation data. For example, it’s possible that participants may have continued to engage in the positive affect up-regulation strategy throughout the presentation of the subsequent stimulus cue (and thus devoted minimal attention to actually processing the cue), but unfortunately there is no way to discern the degree to which the subsequent cue was processed and the prior affective up-regulation strategy ceased. Given that EMG and SCR data were presumably collected during this run as well, I wonder if the authors can also test to see if EMG and SCR data support the idea that affective up-regulation has a distinct psychophysiological readout and whether EMG and SCR data following Mod-CR trials vs. Mod-PS trials show differences as a function of affective up-regulation?

Areas for clarification

-During the Identification Task, were all IAPS images positive in valence? What were the criteria utilized to select these images?

-It’s also unclear what portions of this task were utilized to train the MVPA decoder. Was it the brain activity during passive viewing of the images, or that during the active recall period, or both? Moreover, how did the authors verify that these images actually induced a state of positive emotion in the participants?

-The scaling of the HRF function in training the MVPA decoder by the valence score rating difference from the mean Likert rating assumes, to some extent, that each individual perceived the image to be as positive or negative as the normed rating provided in the IAPS images. This seems to be a strong assumption given that there is probably a great deal of individual variability in subjective responses to these images. Why was a uniform weighting of the HRF not employed? How much variability was there in the weighting of the HRF within a particular class (e.g., positive and negative)?

-The average prediction accuracy for valence for the full stimulus set, though statistically significantly different from 0.5, doesn’t seem very high (55%). This is concerning given this decoder was utilized to trigger stimulus presentation in the subsequent run, correct?

-The authors also should describe briefly how the “reliable” stimulus set was arrived at, even though it is published elsewhere.

-Regarding the test of positive affect up-regulation on subsequent neural processing of affective images, did the authors test for an interaction effect by valence of the subsequent affective image? Some were positive and some were negative, correct? One might suspect that the effect of positive affect self-induction on the subsequent affective processing of the triggered image would vary as a function of the image valence.

Reviewer #2: In this manuscript, the author applied a novel experimental design to test the emotional processing bias of affect state towards subsequent stimulus. A machine learning model based neural feedback is involved to establish prior positive affective state. Overall, I think the research question is interesting and important to the field and the experimental method is novel. However, I have some doubts on neural feedback part as well as some comments on the main results presentation.

Neural feedback: The effectiveness of point-to-hyperplane distance depends on the performance of the decoding model. In other word, if the decoding model based on individual subject does not perform well, the hyperplane could be randomly generated. Point-to-hyperplane distance is meaningless in this scenario. Table 1 had reported average performance in group level but not in individual level. I wonder if there are any individuals have low decoding performances on affective processing (e.g. not significant for within-subject classification)? If so, what strategy should be applied to them? If not, is there any subject-level correlation between decoding performance and regulation ability? Such correlation could imply the robustness of predefined decoding model is one factor for valence-regulation in current neurofeedback setting.

During reading, I found it was hard to catch corresponding result that demonstrates the main claim: affective state bias emotion processing of subsequent stimulus. None of the main figure presents this effect. Although this point is being raised in result section by showing significant beta and p value for GLMM model, a better explanation of GLMM result is needed. Adding any equations or figures to describe this model will be helpful for readers to better understand this result (e.g adding a figure between figure 5 and 6, or adding a new panel to figure 5 to illustrate how the data fits in model and model performance). Also, is there any analysis done to directly compare neural-feedback guided and unguided explicit affect regulation in order to prove the necessity of neural-feedback?

Some minor issues:

Please provide the full name of GLMM. I assume it’s Generalized linear mixed models, but I didn’t find any place to clarify that

In figure 2 bottom diagram, it might be better to add another line to illustrate the scenario that the real-time valence estimate failed to reach initial threshold (0.8) but reach the reduced threshold.

6. PLOS authors have the option to publish the peer review history of their article (what does this mean?). If published, this will include your full peer review and any attached files.

Reviewer #1: No

Reviewer #2: No

---

## [Author Response · Author response to Decision Letter 0]

28 Jan 2022

RESPONSE TO REVIEWERS

We appreciate the reviewers’ thoughtful comments on our prior submission. In response we have added new data analyses and clarifications that have resulted in a much improved manuscript. We first describe author initiated changes that we made to the manuscript, which we deemed necessary based on detection of small omissions or errors in our prior submission. We then reiterate each reviewer’s comment or concern below and describe our specific response.

Author Initiated Changes:

1. We altered Fig 2 to include example image stimuli for which the authors own the copyrights.

2. We incorrectly reported the template space of Fig 4 as Talairach space. We have corrected this to report the space as MNI. 

3. We altered Fig 7 (previously Fig 6) to plot the axes overtop of the individual data markers. In the original version of the figure the data markers occluded parts of the x- and y-axes.

4. We replaced references to the F-test with t-test throughout the manuscript. We originally reported the statistical tests for Matlab’s lme function as F-tests. This is incorrect. Matlab’s lme function uses the t-test to calculate significance of fixed effects. However, the p-values and effect sizes reported in the previous manuscript were correct.

5. We modified the description of our use of fmriprep software in the Main Manuscript and modified the S1 Methods to include fmriprep’s detailed description of the processing pipeline, which the authors of the tool request be included verbatim in all publications (the text is released under the [CC0](https://creativecommons.org/publicdomain/zero/1.0/) license.

Responses to Reviewer #1:

This is a technically impressive investigation of the potential for fMRI to decode affective states from pictorial images and to utilize this information to track and trigger subsequent presentation of affective stimuli.

I have several major comments and then some additional areas for clarification:

1). One conceptual issue I note as problematic in the presentation and framing of these results is that the design and execution of this task somehow replicates or captures key ingredients of CBT. With some notable exceptions (e.g. reminiscence therapy), CBT does not require or engage the participant in deliberate up-regulation of positive emotion via internally-generated strategies, but it instead focuses on realistic appraisal of thoughts and situations with the typically focus on removing or altering distorted interpretations that tend to support and generate negative affect. This is the typical form of cognitive reappraisal as practiced in numerous CBT treatments, and it is not about “positive thinking” but “realistic thinking.” The approach utilized here is qualitatively different, and I think this needs to be better explained in the Introduction and Discussion.

In response, we have removed references to CBT throughout the manuscript (Introduction paragraph: lines 51-59, 68-76 and Discussion paragraph: lines 601-613). We have recast the contributions of this work in terms of attentional deployment strategies as conceptualized by the Process Model of emotion regulation.

2). The use of the word “causal” in the title and throughout the manuscript is too provocative and controversial a term and does not accurately reflect the approach utilized in this study. There are numerous arguments about how causality can be inferred and what constitutes causal inference, but I believe the authors are going too far with the use of this term given the experimental design utilized here. This term should be removed in favor of more accurate descriptions such as “experimental manipulation.”

In response, we have removed the word “causal” from the manuscript, S1 Methods, and Open Science Framework repository.

3). Regarding the claim that self-induced positive affective states positively bias the affect processing of subsequent image stimuli, I think there should be more consideration and nuance with this interpretation. First, the coarse time resolution of the fMRI protocol utilized here (and the slow hemodynamic response function more generally) doesn’t allow for a clean separation of carryover effects form the positive affect up-regulation period to the processing of the triggered stimulus cue, and it’s not clear how these two periods interact in the resultant brain activation data. For example, it’s possible that participants may have continued to engage in the positive affect up-regulation strategy throughout the presentation of the subsequent stimulus cue (and thus devoted minimal attention to actually processing the cue), but unfortunately there is no way to discern the degree to which the subsequent cue was processed and the prior affective up-regulation strategy ceased. Given that EMG and SCR data were presumably collected during this run as well, I wonder if the authors can also test to see if EMG and SCR data support the idea that affective up-regulation has a distinct psychophysiological readout and whether EMG and SCR data following Mod-CR trials vs. Mod-PS trials show differences as a function of affective up-regulation?

We appreciate this comment and recognize this as a critical question of the approach. In response we want to clarify what we think was being conveyed in the comment above. We believe that the reviewer requests an independent test, using psychophysiology, of the two separate parts of the Mod-FS trials. The reviewer wants a psychophysiology test of the volitional self-induction step of the trial (Fig 2, Mod-FS: self-induce) as well as a psychophysiological test of the main experimental manipulation, i.e., differential psychophysiological responses to image-based induction of the feedback-triggered stimuli versus passive stimuli.

A quick note on our reasoning for where and when we used psychophysiology to independently validate neuroimaging findings in this experiment. Our group has published multiple papers on the relative sensitivity of neural decodings versus psychophysiological responses (SCR, facial EMG, and HR) in detecting affect processing. We have found, repeatedly, that neural decodings are approximately 3 times more sensitive than SCR in detecting arousal[1] and approximately 10 times more sensitive than fEMG or HR deceleration in detecting valence[2,3]. In this manuscript, we used psychophysiology to independently validate the induction of affect processing brain states by image stimuli during the Id-PS trials. We have reproduced these effects multiple times which allows us to know that these manipulations can be detected by psychophysiological responses and act as independent validation of the training dataset on which our neural decodings were constructed. This is a recommended step for validating affect induction within neuroimaging experiments[4]. However, we understand slow temporal evolution of the HRF raises skepticism of any claim relying on relatively fast fMRI dynamics. 

Therefore, in response to this comment, we conducted these requested independent tests. First, using a GLMM we detected significant responses (both SCR and cEMG) to volitional self-induction of valence. This finding is now reported in the Results (lines 484-490). Second, using a GLMM (with similar fixed and random effects as those used for the neural decodings) we detected that the experimental manipulation has a differential effect on SCR but not cEMG. We report these findings in the Results (lines 536-546) and describe our interpretation of these findings in the Discussion (lines 623-626).

An additional step we took to remove concerns surrounding HRF dynamics influencing the experiment was to include the decoding of the trigger (or the previous EPI volume in passive trials) in the GLMM of our main experimental manipulation. Controlling for the potential entrainment of affect processing signal, we still see a robust effect of the primary experimental manipulation (see Fig 5 and lines 495-522).

Areas for clarification

-During the Identification Task, were all IAPS images positive in valence? What were the criteria utilized to select these images?

The answer is, no, neural decoding models were trained on affect processing brain states induced by image stimuli from Id-PS trials which were drawn from a broad range of valence and arousal scores. The specific valence and arousal scores of all stimuli are depicted in Fig 3 (Id-PS stimuli are the gray markers). In response, we have highlighted this fact in the manuscript’s Methods (lines 234-235), which describes the maximal spanning of the arousal-valence plane (more, precisely, a maximal subspace span[1]).

-It’s also unclear what portions of this task were utilized to train the MVPA decoder. Was it the brain activity during passive viewing of the images, or that during the active recall period, or both? Moreover, how did the authors verify that these images actually induced a state of positive emotion in the participants?

Similar to the comment above, the neural decoding models were trained within-subject using the brain states induced by the Id-PS trials. We independently validated that affect processing induction occurs via psychophysiological responses for the independent affective dimensions (SCR for arousal and cEMG for valence). This is the process recommended by Heller et al.[4] for validating affect processing induction in neuroimaging experiments. We describe this validation of affect induction in Section: Psychophysiological Response Validation of Affect Processing Induction via Image Stimuli (lines 363-383).

-The scaling of the HRF function in training the MVPA decoder by the valence score rating difference from the mean Likert rating assumes, to some extent, that each individual perceived the image to be as positive or negative as the normed rating provided in the IAPS images. This seems to be a strong assumption given that there is probably a great deal of individual variability in subjective responses to these images. Why was a uniform weighting of the HRF not employed? How much variability was there in the weighting of the HRF within a particular class (e.g., positive and negative)?

We acknowledge that there exists individual variability in affect processing responses to the IAPS stimuli. However, there is strong evidence in the literature that, for classification purposes, the assumption that individuals respond congruently to the normative scores (when discretized according to the middle Likert score). These effects have been reproduced by multiple labs for both within-subject and between-subject neural decoding experiments[1,2,5–7]. Our group also has published a study comparing neural decoding performance when the MVPA is trained with normative scores versus self-reported scores - no significant differences were found in classification performance8.

With respect to the variable weighting, this technique is used to simulate activation-label relationships of the beta-series within the real-time decoding process. We cannot truly construct beta-series in real-time due the long tail of the HRF existing in the future (i.e., the volumes necessary for the regression haven’t been acquired yet). Rather, real-time MVPA decodes each volume individually as the arrive from the reconstruction computer. In training the real-time decoder, however, we can shape the labels of the training set to stimulate the fluctuations that would be observed and to approximate some of the benefits of beta-series in real-time. These variable weightings are specific to the low-quality real-time decoding models and were not used in the high-quality post-hoc decoding models. 

-The average prediction accuracy for valence for the full stimulus set, though statistically significantly different from 0.5, doesn’t seem very high (55%). This is concerning given this decoder was utilized to trigger stimulus presentation in the subsequent run, correct?

Decoding performance is largely driven by the difficulty of the underlying problem. Neural decoding models of affect and emotion processing are built from brain states that are induced by stimuli. However, experimenters control the distribution of the properties of these stimuli. Past decoding modeling efforts[6,7] used hand-chosen image stimuli that clustered into extremes of affective and emotional experience, thereby rendering the underlying decoding problem easier to classify, resulting in high classification accuracy. However, these stimulus sets do not generalize to ecological affective/emotional experience. Our group has published multiple papers[2,5,8] that have explored decoding of computationally sampled stimuli that reflect the maximum range of affective experiences that can be induced by the IAPS image set (see Fig 3, Id-PS stimuli). We have also devised and validated an algorithm for identifying stimuli that reliably induce similar affective brain states across subjects, which we term the Reliable Stimulus Set (RSS)[5]. We have shown that RSS stimuli cluster at the extremes of perceived affective experience (both valence and arousal) and resemble the hand-chosen stimulus sets used to report affect/emotion decoding performance in the literature. Decoding models achieve very high classification performance on RSS stimuli compared to stimuli drawn from the full affective range of experience within IAPS. In fact, our decoding approach achieves accuracies of .75-.79 on RSS, which is similar to the best reported classification performance in the literature[5,6]. In this context, the statistically significant performance (accuracies of .55-.61) we report for the full stimulus set, a much more challenging and ecologically relevant stimulus set, is strong.

We feel it is important to report decoding performance on the full stimulus set, which represents the expected out-of-sample real-world performance of the model, as well as decoding performance that matches results found in the literature, which we represent as the performance decoding the RSS (a subset of the full stimulus set). However, as we have reported on these findings multiple times in the past (including independent psychophysiological confirmation of these effects)[1,2,5] we do not feel these results are novel and should not be included in the main manuscript as primary findings. Therefore, we have reported extensive details on our stimulus selection approach, the role of the RSS, and the validation of our decoding methods in the (supplemental) S1 Methods. We have revised our summary of these findings in the Results section of the main manuscript to reflect this emphasis.

-The authors also should describe briefly how the “reliable” stimulus set was arrived at, even though it is published elsewhere.

We include a response to this comment as part of the response to the comment above.

-Regarding the test of positive affect up-regulation on subsequent neural processing of affective images, did the authors test for an interaction effect by valence of the subsequent affective image? Some were positive and some were negative, correct? One might suspect that the effect of positive affect self-induction on the subsequent affective processing of the triggered image would vary as a function of the image valence.

This is an excellent point, which we had not originally considered. In response, we included a fixed effect for the interaction between the main experimental manipulation (feedback triggered stimulus vs passive) and normative valence score of the subsequent stimulus, which was not significant (see Fig 6). We updated our manuscript to reflect this finding (line 494 to line 521). We included similar interaction terms in our psychophysiological confirmation of the effects of the main experimental manipulation (see line 536 to line 546).

Responses to Reviewer #2:

 In this manuscript, the author applied a novel experimental design to test the emotional processing bias of affect state towards subsequent stimulus. A machine learning model based neural feedback is involved to establish prior positive affective state. Overall, I think the research question is interesting and important to the field and the experimental method is novel. However, I have some doubts on neural feedback part as well as some comments on the main results presentation.

Neural feedback: The effectiveness of point-to-hyperplane distance depends on the performance of the decoding model. In other word, if the decoding model based on individual subject does not perform well, the hyperplane could be randomly generated. Point-to-hyperplane distance is meaningless in this scenario. Table 1 had reported average performance in group level but not in individual level. I wonder if there are any individuals have low decoding performances on affective processing (e.g. not significant for within-subject classification)? If so, what strategy should be applied to them? If not, is there any subject-level correlation between decoding performance and regulation ability? Such correlation could imply the robustness of predefined decoding model is one factor for valence-regulation in current neurofeedback setting.

In response, we conducted this analysis and detected a small but significant effect (see S2 Fig) of model performance positively associating with the magnitude of valence at triggering. This effect matches the concern described above. Therefore, we recomputed the GLMM that modeled our primary experimental manipulation controlling for within-subject model accuracy as a fixed-effect. Here, the effect was not significant. We have revised the manuscript to reflect these changes (line 494 to line 521). 

Also in response to this comment we reported the number of subjects who achieve within-subject significant decoding performance (34/39). We have revised the manuscript to report these findings (line 386 to line 400).

During reading, I found it was hard to catch corresponding result that demonstrates the main claim: affective state bias emotion processing of subsequent stimulus. None of the main figure presents this effect. Although this point is being raised in result section by showing significant beta and p value for GLMM model, a better explanation of GLMM result is needed. Adding any equations or figures to describe this model will be helpful for readers to better understand this result (e.g adding a figure between figure 5 and 6, or adding a new panel to figure 5 to illustrate how the data fits in model and model performance). Also, is there any analysis done to directly compare neural-feedback guided and unguided explicit affect regulation in order to prove the necessity of neural-feedback?

In response to the first point, we constructed a new figure (see Fig 6) that summarizes these effects visually.

In response to the second point, no, comparing real-time guided to unguided self-induction was not part of the experiment design. This is a relatively small-scale neuroimaging study (funded by a NARSAD YIA award with a small budget). The intent was to maximize the effect size of the experimental manipulation using as few subjects as possible. This is an interesting question - whether the feedback or the dynamic trigger is the most critical element of the technology. However, that question is beyond the scope of this work.

Some minor issues:

Please provide the full name of GLMM. I assume it’s Generalized linear mixed models, but I didn’t find any place to clarify that

In response, we have modified the manuscript to provide the full name of the Generalized linear mixed-effects model (GLMM) at first usage (line 369) in the manuscript before switching to the acronym for the remainder of the text. 

In figure 2 bottom diagram, it might be better to add another line to illustrate the scenario that the real-time valence estimate failed to reach initial threshold (0.8) but reach the reduced threshold.

In response, we have elaborated the caption of Figure 2 to more clearly describe the dashed line representing the Trigger threshold as a boundary that evolves through time. We chose this solution in order to maintain the one-to-one relationship between the elements of the trial (denoted by the image blocks) and the diagram of the real-time triggering system. A second line would not allow for this one-to-one correspondence.

References

1. Bush, K. A., Privratsky, A., Gardner, J., Zielinski, M. J. & Kilts, C. D. Common Functional Brain States Encode both Perceived Emotion and the Psychophysiological Response to Affective Stimuli. Scientific Reports 8, (2018).

2. Wilson, K. A., James, G. A., Kilts, C. D. & Bush, K. A. Combining Physiological and Neuroimaging Measures to Predict Affect Processing Induced by Affectively Valent Image Stimuli. Sci Rep 10, 9298 (2020).

3. Bush, K. A., James, G. A., Privratsky, A. A., Fialkowski, K. P. & Kilts, C. D. An action-value model explains the role of the dorsal anterior cingulate cortex in performance monitoring during affect regulation. bioRxiv 23 (2020) doi:10.1101/2020.09.08.283671.

4. Heller, A. S., Greischar, L. L., Honor, A., Anderle, M. J. & Davidson, R. J. Simultaneous acquisition of corrugator electromyography and functional magnetic resonance imaging: A new method for objectively measuring affect and neural activity concurrently. NeuroImage 58, 930–934 (2011).

5. Bush, K. A. et al. Brain States That Encode Perceived Emotion Are Reproducible but Their Classification Accuracy Is Stimulus-Dependent. Frontiers in Human Neuroscience 12, (2018).

6. Baucom, L. B., Wedell, D. H., Wang, J., Blitzer, D. N. & Shinkareva, S. V. Decoding the neural representation of affective states. NeuroImage 59, 718–727 (2012).

7. Chang, L. J., Gianaros, P. J., Manuck, S. B., Krishnan, A. & Wager, T. D. A Sensitive and Specific Neural Signature for Picture-Induced Negative Affect. PLOS Biology 13, e1002180 (2015).

8. Bush, K. A., Inman, C. S., Hamann, S., Kilts, C. D. & James, G. A. Distributed Neural Processing Predictors of Multi-dimensional Properties of Affect. Frontiers in Human Neuroscience 11, (2017).

---

## [Decision Letter · Decision Letter 1]

17 Feb 2022

A test of affect processing bias in response to affect regulation

PONE-D-21-19149R1

Dear Dr. Bush,

We’re pleased to inform you that your manuscript has been judged scientifically suitable for publication and will be formally accepted for publication once it meets all outstanding technical requirements.

Kind regards,

Desmond J. Oathes

Academic Editor

PLOS ONE

Additional Editor Comments (optional):

Reviewers' comments:

Reviewer's Responses to Questions

**Comments to the Author**

1. If the authors have adequately addressed your comments raised in a previous round of review and you feel that this manuscript is now acceptable for publication, you may indicate that here to bypass the “Comments to the Author” section, enter your conflict of interest statement in the “Confidential to Editor” section, and submit your "Accept" recommendation.

Reviewer #1: All comments have been addressed

Reviewer #2: All comments have been addressed

2. Is the manuscript technically sound, and do the data support the conclusions?

Reviewer #1: Yes

Reviewer #2: Yes

3. Has the statistical analysis been performed appropriately and rigorously? 

Reviewer #1: Yes

Reviewer #2: Yes

4. Have the authors made all data underlying the findings in their manuscript fully available?

Reviewer #1: Yes

Reviewer #2: Yes

5. Is the manuscript presented in an intelligible fashion and written in standard English?

Reviewer #1: Yes

Reviewer #2: Yes

6. Review Comments to the Author

Reviewer #1: (No Response)

Reviewer #2: Glad to see the concern regarding within-participants decoder performance is addressed. According to this data, it seems that there is correlation between decoder performance and magnitude of valence at triggering. but it does not influence the main effect of this experiment. This could be an important note to help researchers who plan to follow this novel paradigm.

The new figure 6 is informative and presents the main results well. I think the figures and analysis are in good shape and fit well with main claim. I have no further concern regarding this manuscript.

7. PLOS authors have the option to publish the peer review history of their article (what does this mean?). If published, this will include your full peer review and any attached files.

Reviewer #1: No

Reviewer #2: **Yes: **Ke Bo

---

## [Editor Report · Acceptance letter]

21 Feb 2022

PONE-D-21-19149R1 

A test of affect processing bias in response to affect regulation 

Dear Dr. Bush:

I'm pleased to inform you that your manuscript has been deemed suitable for publication in PLOS ONE. Congratulations! Your manuscript is now with our production department. 

Kind regards, 

on behalf of

Dr. Desmond J. Oathes 

Academic Editor

PLOS ONE